# A gene-expression-based neural code for food abundance that modulates lifespan

Eugeni V Entchev[1†], Dhaval S Patel[1†], Mei Zhan[2,3†], Andrew J Steele[1], Hang Lu[2,3,4*], QueeLim Ch'ng[1*]

[1]MRC Centre for Developmental Neurobiology, King's College London, London, United Kingdom; [2]Interdisciplinary Bioengineering Graduate Program, Georgia Institute of Technology, Atlanta, United States; [3]Wallace H. Coulter Department of Biomedical Engineering, Georgia Institute of Technology, Atlanta, United States; [4]School of Chemical and Biomolecular Engineering, Georgia Institute of Technology, Atlanta, United States

**Abstract** How the nervous system internally represents environmental food availability is poorly understood. Here, we show that quantitative information about food abundance is encoded by combinatorial neuron-specific gene-expression of conserved TGFβ and serotonin pathway components in *Caenorhabditis elegans*. Crosstalk and auto-regulation between these pathways alters the shape, dynamic range, and population variance of the gene-expression responses of *daf-7* (TGFβ) and *tph-1* (tryptophan hydroxylase) to food availability. These intricate regulatory features provide distinct mechanisms for TGFβ and serotonin signaling to tune the accuracy of this multi-neuron code: *daf-7* primarily regulates gene-expression variability, while *tph-1* primarily regulates the dynamic range of gene-expression responses. This code is functional because *daf-7* and *tph-1* mutations bidirectionally attenuate food level-dependent changes in lifespan. Our results reveal a neural code for food abundance and demonstrate that gene expression serves as an additional layer of information processing in the nervous system to control long-term physiology.

*For correspondence: hang.lu@gatech.edu (HL); queelim@kcl.ac.uk (QC)

†These authors contributed equally to this work

Competing interests: The authors declare that no competing interests exist.

## Introduction

All organisms need to accurately assess their environment to respond to changes that impact their survival. Environmental changes such as food availability can lead to alterations in organismal physiology, such as stress resistance and metabolic states that have consequences for clinically important outputs such as disease progression, health, fecundity and lifespan (*Libert and Pletcher, 2007*). Many conserved genetic mechanisms that govern these alterations to physiology have been identified (*Libert and Pletcher, 2007*; *Berthoud and Morrison, 2008*; *Rother et al., 2008*; *Alcedo et al., 2010*; *Koch and Horvath, 2014*). Yet, how these genetic pathways encode and process information about the environment to elicit physiological outputs in vivo is unclear at a quantitative and mechanistic level, despite their importance for health and disease.

In animals, the nervous system is the central site for processing sensory information and coordinates organism-wide responses to changing conditions. Food availability is a critical environmental variable that modulates metabolism and other physiological outputs via neuroendocrine circuits (*Berthoud and Morrison, 2008*; *Rother et al., 2008*; *Alcedo et al., 2010*; *Koch and Horvath, 2014*). In contrast to well-studied sensory modalities such as vision and olfaction (*Baier, 2013*; *Wilson, 2013*), where neural processing occurs on short timescales using electrical signals, how food availability is internally represented across a broad range of inputs to regulate long-term, food-related physiological responses remains virtually unknown. A particularly interesting food-related response is the role of dietary restriction (DR) in modulating lifespan in diverse species (*Bishop and*

**eLife digest** To maximize their chances of survival, animals need to be able to sense changes in the abundance of food in their environment and respond in an appropriate manner. The nervous system is able to sense cues from the environment and coordinate responses in the whole organism, but it is not clear how this leads to long-term changes in the organism's biology.

In nematode worms, two genes called *daf-7* and *tph-1* appear to be involved in connecting the sensing of food availability with changes in the biology of the organism. The *daf-7* gene encodes a hormone, while *tph-1* encodes an enzyme that makes a neurochemical called serotonin.

Here, Entchev, Patel, Zhan et al. found that *daf-7* and *tph-1* genes are active in three pairs of neurons in nematode worms. The experiments show that these neurons collectively form a circuit that carries information about the abundance of food, which leads to changes in how long the worms live. When this circuit was disrupted by removing these genes, the worms' ability to adjust their lifespan in response to changes in the availability of food was weakened, likely because they were unable to sense food. The experiments also show that the circuit regulates itself, largely because *daf-7* and *tph-1* are able to control each-other's activity.

Together, these results suggest that changing the activity of certain genes in neurons enables nematode worms to alter their biology in response to changes in the availability of food. Neurons in the brain use electrical activity to communicate and process information and Entchev, Patel, Zhan et al.'s findings imply that gene activity can also perform a similar role.

*Guarente, 2007*; *Mair and Dillin, 2008*; *Fontana et al., 2010*; *Alic and Partridge, 2011*). DR occurs through changes that likely happen over long timescales (hours to days), unlike fast behavioral responses to visual or olfactory cues. Neural gene expression also occurs over long timescales (minutes to hours) and is thus well suited for functionally encoding food abundance during DR.

In *Caenorhabditis elegans*, *daf-7* and *tph-1* are conserved components of neural TGFβ and serotonin signaling pathways, respectively, and are associated with food sensing and modulation of organismal physiology. *daf-7* encodes a TGFβ family member (*Ren et al., 1996*), while *tph-1* encodes tryptophan hydroxylase, the rate-limiting enzyme for serotonin synthesis (*Sze et al., 2000*). In *C. elegans*, TGFβ and serotonin signaling affect lifespan and metabolism, consistent with conserved roles from invertebrates to mammals (*Sze et al., 2000*; *Ashrafi, 2007*; *Murakami and Murakami, 2007*; *Petrascheck et al., 2007*; *Shaw et al., 2007*; *Brown and Schneyer, 2010*; *Oury and Karsenty, 2011*). *tph-1* and *daf-7* are expressed in an environmentally responsive manner in specific neurons with food-related functions (*Ren et al., 1996*; *Schackwitz et al., 1996*; *Sze et al., 2000*; *Zhang et al., 2005*; *Chang et al., 2006*; *Liang et al., 2006*; *Greer et al., 2008*; *Pocock and Hobert, 2010*). *daf-7* is expressed in the ASI sensory neurons, whose activities are responsive to bacterial food (*Ren et al., 1996*; *Gallagher et al., 2013*; *Zaslaver et al., 2015*). Starvation reduces *daf-7* expression in ASI, and laser ablations of ASI extend lifespan, consistent with the role of *daf-7* and other ASI-expressed genes in modulating lifespan (*Ren et al., 1996*; *Alcedo and Kenyon, 2004*; *Bishop and Guarente, 2007*). *tph-1* is expressed in the NSM foregut neurons, the ADF sensory neurons, and the HSN motorneurons involved in egg-laying (*Sze et al., 2000*). Both serotonin signaling mutants and NSM ablation affect food-modulated locomotion, consistent with the idea that serotonin from NSM acts in this food-related response (*Sawin et al., 2000*). In the food-responsive ADF neurons (*Zaslaver et al., 2015*), *tph-1* expression is responsive to pathogenic bacteria and starvation, to respectively mediate aversive olfactory plasticity and stress responses (*Zhang et al., 2005*; *Liang et al., 2006*).

*daf-7* and *tph-1* are therefore strong candidates for mediating the link between environmental cues and longevity. Nonetheless, how these genes cooperate to quantitatively encode a broad range of food levels to modulate lifespan is unknown. Gene-expression responses to food cues have largely been studied as ON/OFF switches to the presence or absence of food (*Zinke et al., 2002*; *Baugh et al., 2009*). Because food abundance is a continuous variable, we sought to understand how expression of *tph-1* and *daf-7* could allow animals to distinguish multiple food levels. Furthermore, gene expression is inherently variable (*Eldar and Elowitz, 2010*), but this property is rarely studied in vivo in multicellular animals; thus we also sought to determine how gene-expression variability affects the ability of the worm to encode its environment.

Here we show that *daf-7* and *tph-1* expression in three pairs of neurons forms a distributed circuit that quantitatively encodes food abundance and mediates dietary effects on lifespan in *C. elegans*. Specific disruptions to this circuit resulted in corresponding attenuation in the ability to discriminate between food levels in both the gene-expression code and lifespan output. We found that this circuit tunes its own accuracy, largely via the regulation of the dynamic range and variability of food-responsive gene expression by *tph-1* and *daf-7* signalling, respectively. Our work suggests that neural regulation of gene expression in conserved pathways can couple environmental sensation to physiological output, and highlights a novel mechanism for information processing by the nervous system to impact physiology.

## Results

### Two neuronal genes mediate bidirectional effects of DR on lifespan

During DR, lifespan increases as food levels are decreased from *ad libitum* conditions until reaching a maximum, beyond which further food reduction lowers lifespan (*Bishop and Guarente, 2007*; *Mair and Dillin, 2008*; *Fontana et al., 2010*; *Alic and Partridge, 2011*). To fully understand the response to food levels that *C. elegans* might encounter in the wild (*Felix and Duveau, 2012*), we modified a well-established DR protocol (*Greer et al., 2007*) (*Figure 1A*) to measure the lifespans of wildtype animals shifted as day 2 adults to 19 concentrations of the *Escherichia coli* food source across ~11 orders of magnitude (*Figure 1B*, *Figure 1—figure supplement 1* and *Figure 1—source data 1*). We inhibited progeny production with *egg-5(RNAi)* (*Figure 1A*) to prevent matricide due to internal hatching at low food levels. This treatment does not affect the lifespan response to food; similar responses were observed in wildtype animals without *egg-5(RNAi)* (*Figure 1—figure supplement 1*), and are found in the literature where similar subsets of food ranges were tested using other DR protocols (below).

We uncovered a multiphasic relationship between bacterial abundance and longevity that is more complex than previously reported with smaller concentration ranges (*Figure 1B*) (*Greer et al., 2007*; *Greer and Brunet, 2009*; *Ching et al., 2010*). We found that lifespan increased and then decreased as bacterial concentration was reduced from the highest level, forming a DR response consistent with prior reports at high food ranges (*Bishop and Guarente, 2007*; *Panowski et al., 2007*; *Greer and Brunet, 2009*; *Mair et al., 2009*; *Ching et al., 2010*). Surprisingly, upon further reduction, lifespan increased again till a plateau was reached, suggesting that the initial decrease was not due to limiting nutrients. The longest lifespans occurred in the absence of bacteria, where the magnitudes of these effects were consistent with published dietary deprivation experiments (*Kaeberlein et al., 2006*; *Lee et al., 2006*). This relationship between lifespan and food abundance was maintained across temperatures (*Figure 1—figure supplement 1*), suggesting a robust food-sensing process. This multiphasic food response may reflect trade-offs between multiple food-regulated processes as previously discussed (*Kaeberlein et al., 2006*; *Lee et al., 2006*). Here we used the complex lifespan response as a functional basis for understanding how neuronal gene expression could encode food abundance.

To understand how the multiphasic lifespan response to food abundance is regulated, we measured the lifespan of *daf-7* and *tph-1* null mutants across six bacterial concentrations that captured the complexity of broad-range DR (circled in *Figure 1B*). Prior studies suggested that *daf-7* and *tph-1* mediate lifespan extension (*Murakami and Murakami, 2007*; *Shaw et al., 2007*; *van der Goot et al., 2012*). We showed that their effects were in fact bidirectional: these genes could either extend or reduce lifespan in a food-specific manner (*Figure 1C*, *Figure 1—source data 2, 3*). Both single mutants had reduced lifespans at low food levels and increased lifespan at $6 \times 10^8$ cells/ml in comparison to wildtype; additionally, *daf-7(−)* mutants were long-lived at the highest food level (*Figure 1C*). The magnitude of lifespan changes we observed at high food levels ($1 \times 10^{10}$ cells/ml) were comparable to prior studies performed at *ad libitum* food conditions (*Murakami and Murakami, 2007*; *Shaw et al., 2007*). Intriguingly, *tph-1* and *daf-7* influenced the longevity response more strongly at low and high bacterial concentrations respectively (*Figure 1D*), suggesting that they act at different but overlapping ranges of food. Furthermore, the double mutant resembled the *tph-1(−)* mutant at low bacterial levels and the *daf-7(−)* mutant at high bacterial levels (*Figure 1D*), suggesting that these genes act in parallel rather than in a linear pathway. Together, these phenotypes indicate that *daf-7(−)* and *tph-1(−)* mutants were neither intrinsically long- nor short-lived; instead, their phenotypes and genetic interactions were modulated by extensive gene-environment interactions.

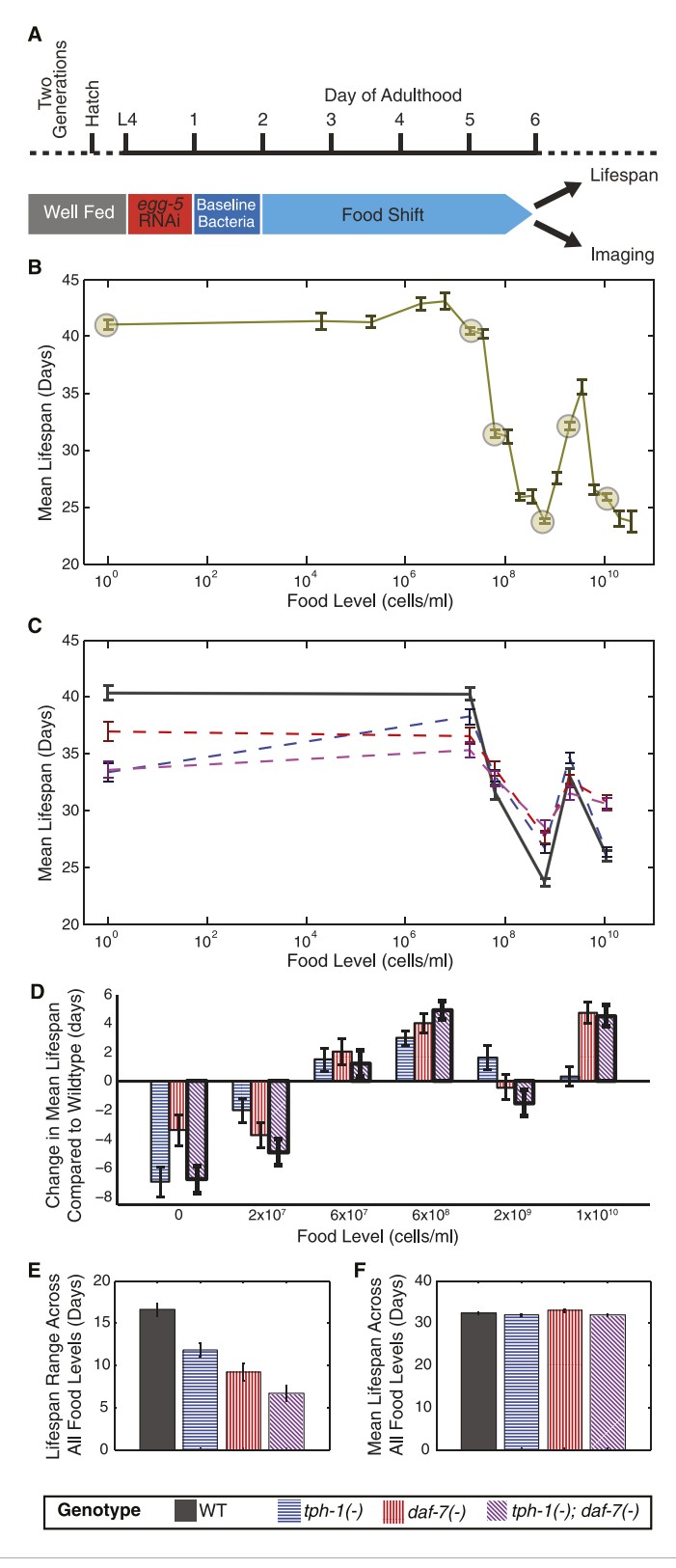

**Figure 1**. Two neuronal genes, *daf-7* and *tph-1*, shape a complex, multiphasic relationship between lifespan and food availability. (**A**) Protocol for maintaining animals at different food levels for lifespan and imaging experiments. Effects of initiating different dietary restriction (DR) on different days are shown in *Figure 1—figure supplement 1A,B*. (**B**) Mean
*Figure 1. continued on next page*

*Figure 1. Continued*

lifespan of wildtype worms subjected to 19 food levels ranging from 0–3.5 × 10$^{10}$ bacterial cells/ml at 20°C. Points denoting key features in the food response and used as food conditions in subsequent experiments are highlighted. *Figure 1—figure supplement 1D* shows that these lifespan responses have similar shapes across different temperatures. The lifespan data are shown in *Figure 1—source data 1*. (**C**) Mean lifespans of wildtype and mutant animals across the six food levels indicated in (**B**) show that loss of *tph-1* and *daf-7* preserves the pattern but attenuates the range of the lifespan response. Genotypes are indicated by the legends below (**E**) and (**F**). The lifespan data are shown in *Figure 1—source data 2*, and statistical comparisons between the different genotypes and food levels are shown in *Figure 1—source data 3*. (**D**) *tph-1* and *daf-7* modulation of lifespan is bidirectional and their epistatic relationship is food-specific. The epistatic interaction between the two genes in the *tph-1(–)*; *daf-7(–)* double mutant differs depending on food level. The double mutant resembled the *tph-1(–)* single mutant in the absence of a bacterial food source and resembled the *daf-7(–)* single mutant at a high bacterial food concentration. (**E**) Range of food-induced lifespan modulation for each genotype. Range is defined by the difference between the highest and lowest mean lifespan response across the six food levels. (**F**) Average of the mean lifespan responses across all food levels for each genotype reveals a consistent, food-independent baseline lifespan response. The schedule for transferring animals to different conditions in these lifespans are shown in *Figure 1—figure supplement 2*.

The following source data and figure supplements are available for figure 1:

**Source data 1**. Summary of wild type lifespan outputs under the full range of food levels tested.

**Source data 2**. Summary of wild type and mutant lifespan outputs under six selected food levels.

**Source data 3**. Statistical significance of lifespan modulation across food levels and genetic backgrounds.

**Figure supplement 1**. Effects of DR initiation time, temperature and *egg-5(RNAi)* on the lifespan response.

**Figure supplement 2**. Schedule of transfers for lifespans at different temperatures.

Rather than altering the basic pattern of the lifespan response to food, loss of *tph-1* or *daf-7*, either alone or in combination, dampened food responsiveness by bidirectionally attenuating extension and reduction of lifespan due to DR (*Figure 1C,D*). This effect was manifested in the diminished range of lifespans across all food levels in both the *daf-7(–)* and *tph-1(–)* single mutants, which was further reduced in the double mutant (*Figure 1E*). This result also supports the idea that these genes act in parallel pathways. Furthermore, the mean lifespan across all food levels were similar in all the genotypes tested (*Figure 1F*), suggesting that mutations in *tph-1* and *daf-7* lowered the food-responsive component of longevity around a consistent, food-independent lifespan that may be specified by other environmental parameters such as temperature (*Figure 1E* and *Figure 1—figure supplement 1*). This bidirectional dampening of the food response and preservation of an underlying lifespan differs from previously described DR regulators, such as *aak-2*, *daf-16*, *pha-4* and *skn-1*, whose mutants abolish DR-mediated lifespan extension (*Bishop and Guarente, 2007*; *Greer et al., 2007*; *Panowski et al., 2007*). Thus, *tph-1* and *daf-7* mutants reveal a previously unobserved DR phenotype, and our results suggest that these genes mediate a bidirectional lifespan response to DR.

## Neuronal expression of *daf-7* and *tph-1* encodes food abundance

*tph-1* is expressed in the ADF sensory neurons, the NSM neurons within the foregut, and the hermaphrodite-specific HSN motor neurons (*Sze et al., 2000*). *daf-7* is expressed in a single pair of ASI sensory neurons (*Ren et al., 1996*; *Schackwitz et al., 1996*). To determine whether *tph-1* and *daf-7* act in these neurons to modulate lifespan, we expressed these genes in specific neurons and tested their ability to rescue the lifespan phenotypes in the *tph-1(–)*; *daf-7(–)* double mutant. Expression of *tph-1* in either ADF or NSM neurons (*Figure 2A–C*) or of *daf-7* in ASI neurons (*Figure 2D,E*) could rescue the lifespan phenotypes at low and high food levels, respectively. These results indicate that the activity of *tph-1* and *daf-7* in these respective neurons are relevant to lifespan modulation.

Previous studies showed that *daf-7* and *tph-1* expression are regulated by environmental cues (*Ren et al., 1996*; *Schackwitz et al., 1996*; *Sze et al., 2000*; *Zhang et al., 2005*; *Chang et al., 2006*;

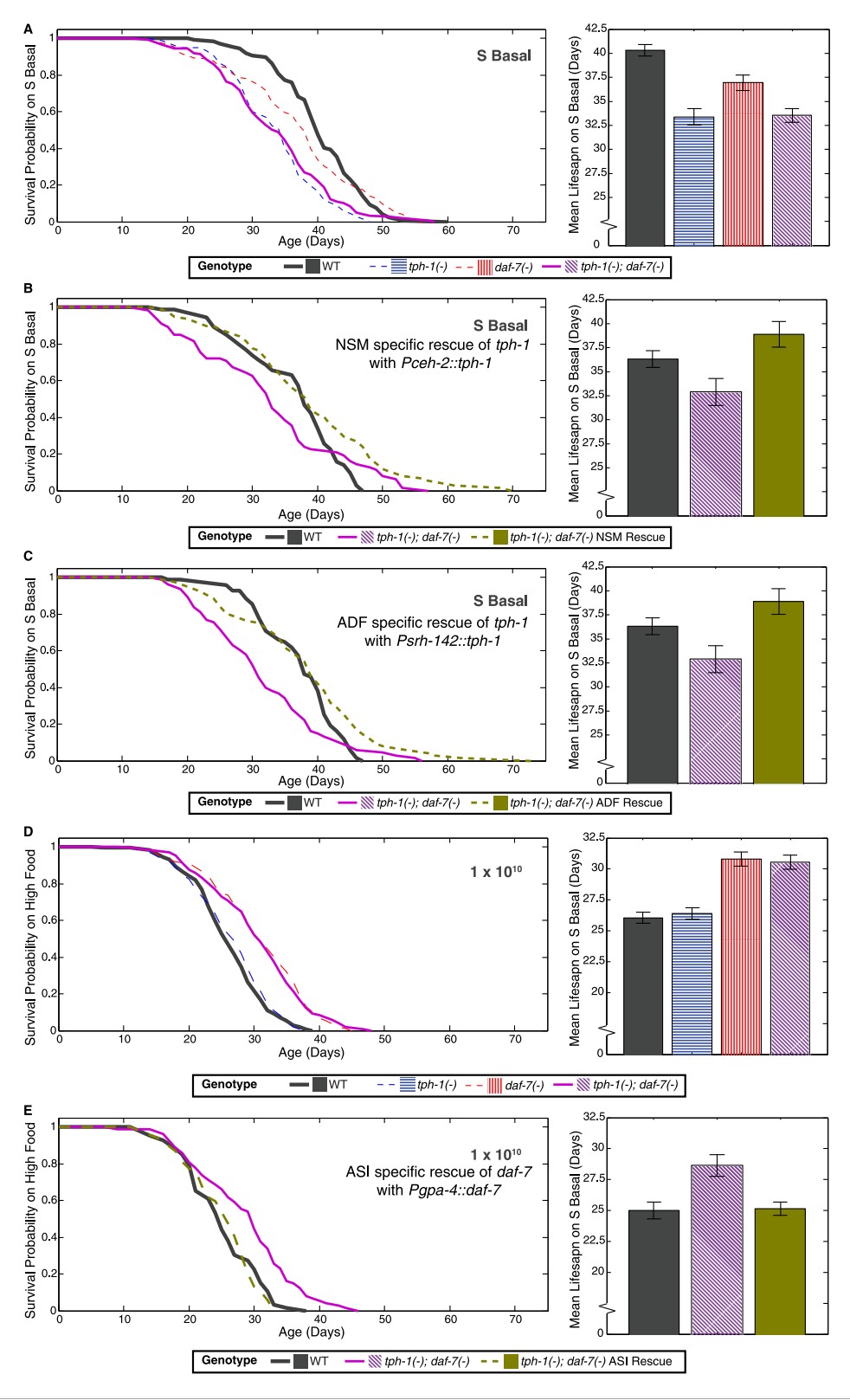

**Figure 2**. Neuron-specific rescue of lifespan phenotypes. (**A**) Lifespan outcomes of wildtype, *tph-1(–)* and *daf-7(–)* single mutants and the *tph-1(–)*; *daf-7(–)* double mutant indicates that the double mutant closely resembles the *tph-1(–)* mutant in the absence of a bacterial food source. (**B**) In the absence of bacterial food, restoration of *tph-1*

*Figure 2. Continued*

activity in the NSM neurons via the expression of a *tph-1* cDNA driven by the *ceh-2* promoter rescues the lifespan reduction observed in the *tph-1(–)*; *daf-7(–)* double mutants. (**C**) Restoration of *tph-1* expression in the ADF neurons via the *srh-142* promoter also shows reversal of the lifespan reduction. (**D**) Lifespan outcomes of wildtype, *tph-1(–)* and *daf-7(–)* single mutants and the *tph-1(–)*; *daf-7(–)* double mutant indicates that the double mutant closely resembles the *daf-7(–)* mutant at a high concentration of the bacterial food source. (**E**) At high food level, restoration of *daf-7* expression in the ASI neurons via the expression of *daf-7* under the *gpa-4* promoter reverses the lifespan extension observed in the *tph-1(–)*; *daf-7(–)* double mutants. All comparisons are drawn against non-transgenic siblings of animals bearing the extrachromosomal array of interest.

*Liang et al., 2006*; *Greer et al., 2008*; *Pocock and Hobert, 2010*). However, their expression profiles over a broad range of inputs remain unknown because manual studies limit the number of animals and environmental conditions that can be feasibly studied in a consistent way. To overcome these limitations, we used an automated, high-throughput microfluidic-based platform (*Figure 3A* and *Figure 3—figure supplement 1*) (*Chung et al., 2008*; *Crane et al., 2012*) for quantitative large-scale imaging of individual worms bearing single-copy fluorescent transcriptional reporters for both *tph-1* and *daf-7* (*Ptph-1::mCherry* and *Pdaf-7::Venus*) across different food levels (*Figure 3B*). For brevity, we refer to these reporter activities as *tph-1* and *daf-7* expression. Our reporters contain the same regulatory regions as published reporters that have been well validated, and show identical expression patterns (*Ren et al., 1996*; *Schackwitz et al., 1996*; *Sze et al., 2000*; *Zhang et al., 2005*; *Chang et al., 2006*; *Liang et al., 2006*; *Greer et al., 2008*; *Pocock and Hobert, 2010*) (*Figure 3B*). Starvation, hypoxia, or pathogenic bacteria alter both *tph-1* reporter expression and serotonin levels (*Zhang et al., 2005*; *Liang et al., 2006*; *Pocock and Hobert, 2010*), while corresponding changes occur in *daf-7* RNA levels and *daf-7* reporter expression (*Ren et al., 1996*). These published results indicate that *tph-1* and *daf-7* reporters are faithful readouts for the expression of their respective genes (see 'Materials and methods' for additional details on reporter validation).

We measured *tph-1* expression levels in both NSM and ADF, and *daf-7* in ASI, in animals exposed to the same six food levels that define our complex DR response (*Figure 3C*). Remarkably, we found that each neuron type had a specific pattern of activity across the six food levels (*Figure 3C*). Even with respect to a single gene, *tph-1*, the expression response in NSM differed from that in ADF, suggesting non-redundant roles of NSM and ADF in encoding bacterial abundance. Consistent with low *tph-1* expression in the absence of food (*Figure 3C*), serotonin levels were reduced in NSM and ADF after starvation ('Materials and methods' and *Liang et al., 2006*). Notably, the responses of *tph-1* in NSM and *daf-7* in ASI were non-monotonic, prohibiting unique representation of the food level using either of these readouts alone. The dynamic range and inter-individual variability of the three expression patterns also differed, suggesting different sensitivities towards the sensory input from food in different neurons (*Figure 3C,D*). Moreover, the distributions of reporter intensity for each neuron type shifted in a graded manner across the six bacterial concentrations (*Figure 3D*) indicating that the expression of *tph-1* and *daf-7* could provide information about a continuous range of environmental inputs for individual animals. This graded response contrasts with many switch-like pathways, such as MAP kinase signalling, that produce outputs with a finite number of (usually two) stable states, where the population responds to changing environmental conditions by shifting sub-populations of cells from one state to another (*Ferrell, 1996*).

Since gene expression may be used to internally represent environmental conditions and mediate long-term physiological outputs such as longevity, we next asked to what extent expression in individual neurons correlates with lifespan across food levels. We found that the average expression levels of the individual genes in each neuron alone were insufficient to either represent food inputs or uniquely specify lifespan outputs (*Figure 4A*), largely due to non-monotonic expression and lifespan responses (*Figures 1, 3C*). However, combining non-monotonic encoding across multiple neurons can increase the representational capability, by allowing animals to use a combinatorial scheme to internally represent environmental conditions with higher resolution (*Figure 4B*).

To assess the accuracy of the internal representation of food levels based on the graded combinatorial expression of *tph-1* and *daf-7* in wildtype animals, we next applied a decoding analysis (*Figure 4C,D*) (*Dayan and Abbott, 2005*). Using only the gene-expression data, we applied this

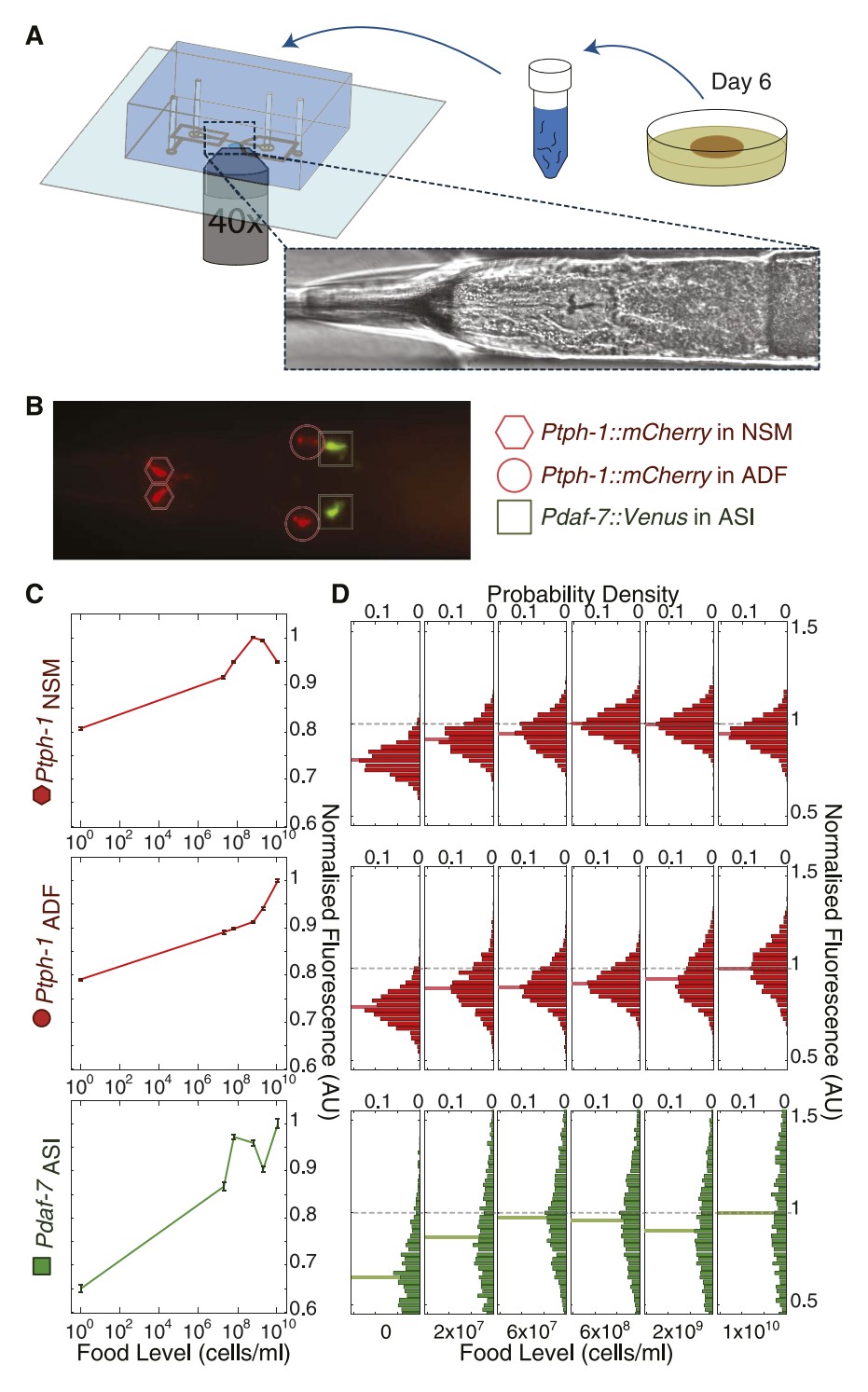

**Figure 3**. High-throughput quantitative imaging of *tph-1* and *daf-7* fluorescent reporters reveals neuron-specific, graded expression responses to food level. (**A**) Microfluidic system enabling high-throughput, neuron-specific quantitative imaging of gene expression in a large number of individual animals. Animals are transferred from culture plates to a liquid suspension at day 6 of adulthood and then loaded into the device for imaging. *Figure 3—figure supplement 1* shows an overview of this imaging system. (**B**) Representative merged fluorescent image of transgenic worm with red and green fluorescent reporters for *tph-1* and *daf-7* transcriptional activity. Shapes indicate locations and identities of specific neurons. (**C**) Mean expression profiles of *tph-1* in NSM

*Figure 3. continued on next page*

*Figure 3. Continued*

(*Ptph-1*NSM) and ADF (*Ptph-1*ADF), and *daf-7* in ASI (*Pdaf-7*ASI) across six different food levels are neuron-specific and largely non-monotonic. Measurements are normalized to the highest mean expression response observed in each respective neuron; error bars are SEM. (**D**) Distribution of the expression responses of *tph-1* in NSM and ADF and *daf-7* in ASI at different food levels. Means are indicated by the solid lighter-shade lines behind the distributions. Dashed line denotes the highest mean expression for each neuron, which was used for normalization.

The following figure supplement is available for figure 3:

**Figure supplement 1**. Overview of the high-throughput quantitative imaging system for single-cell fluorescent intensity measurements in *C. elegans*.

algorithm to infer the food level that individual animals were exposed to. This approach takes into account the entire distribution of gene expression, and thus involves both means and variance to estimate accuracy. Intuitively, decreased overlap between the distributions from different food levels leads to increased accuracy (*Figure 4C*).

We modeled neuronal encoding with a maximum likelihood estimator applied to the population distributions of expression data at each food level. We next inferred food exposure based on expression in individuals and then compared them to the actual food level to determine the accuracy of the gene-expression responses. The results were summarized as matrices showing the frequency of each inferred food stimuli vs the actual food stimuli for each population (*Figure 4C,D*). If the expression distributions were distinct (non-overlapping) across food levels, a correct inference was always obtained, resulting in a strongly diagonalized matrix (*Figure 4C*, top). If the distributions overlapped entirely, then the food level inference was random and the values in the matrix were inversely related to the number of food conditions tested (*Figure 4C*, bottom). Using expression data from NSM, ADF or ASI individually, we showed that each neuronal readout had some discriminatory power (*Figure 4D*). When neuron pairs were combined, and particularly when all three neuron pairs were used, the accuracy improved (*Figure 4D*). This result suggests that non-redundant encoding by each neuron pair improves the accuracy of the overall representation. Generally, this coding system was best at distinguishing the highest and lowest food levels, consistent with the boom and bust lifestyle of *C. elegans* in the wild (*Felix and Duveau, 2012*). Interestingly, the intermediate bacterial level, $6 \times 10^8$ cells/ml, where we observed the lowest lifespan, also showed a heightened accuracy. While the graded and non-monotonic encoding scheme may allow discrimination among certain intermediate food levels, the discriminatory ability was also limited by variation in the responses.

To determine if the accuracy of gene expression was suitable for modulating lifespan, we next assessed how accurately the lifespan distributions could be used to infer the food level using a similar decoding procedure. This analysis is important because the accuracy of the output sets constraints for the accuracy required in the internal representation. Ideally, the accuracy of both the representation and output should be similar, so that the representation carries sufficient information for the output. In this decoding analysis, we assessed whether lifespan could also be used to infer the food level experienced by an individual, using estimators based on the Weibull distribution of our survival data. Remarkably, the inferential accuracy of lifespan was similar to that of gene expression (*Figure 4D*), indicating that neuronal gene expression captured sufficient information about food levels to mediate corresponding lifespan outputs.

## Self- and cross-regulation between *daf-7* and *tph-1* shape gene-expression responses to food

Prior work suggested that *tph-1* and *daf-7* cross-regulate (*Sze et al., 2000*; *Chang et al., 2006*). Thus, we explored the possibility that the internal representation based on *tph-1* and *daf-7* expression in wildtype animals emerges from their mutual regulation. Because our reporter transgenes were distinct from the endogenous genes, we were able to measure reporter expression in *tph-1(–)* and *daf-7(–)* single and double mutants, which could reveal self- and cross-regulatory interactions. These measurements of reporter expression across six food levels uncovered new and extensive regulatory relationships between *tph-1* and *daf-7* that were neuron-specific and food-dependent (*Figure 5A*).

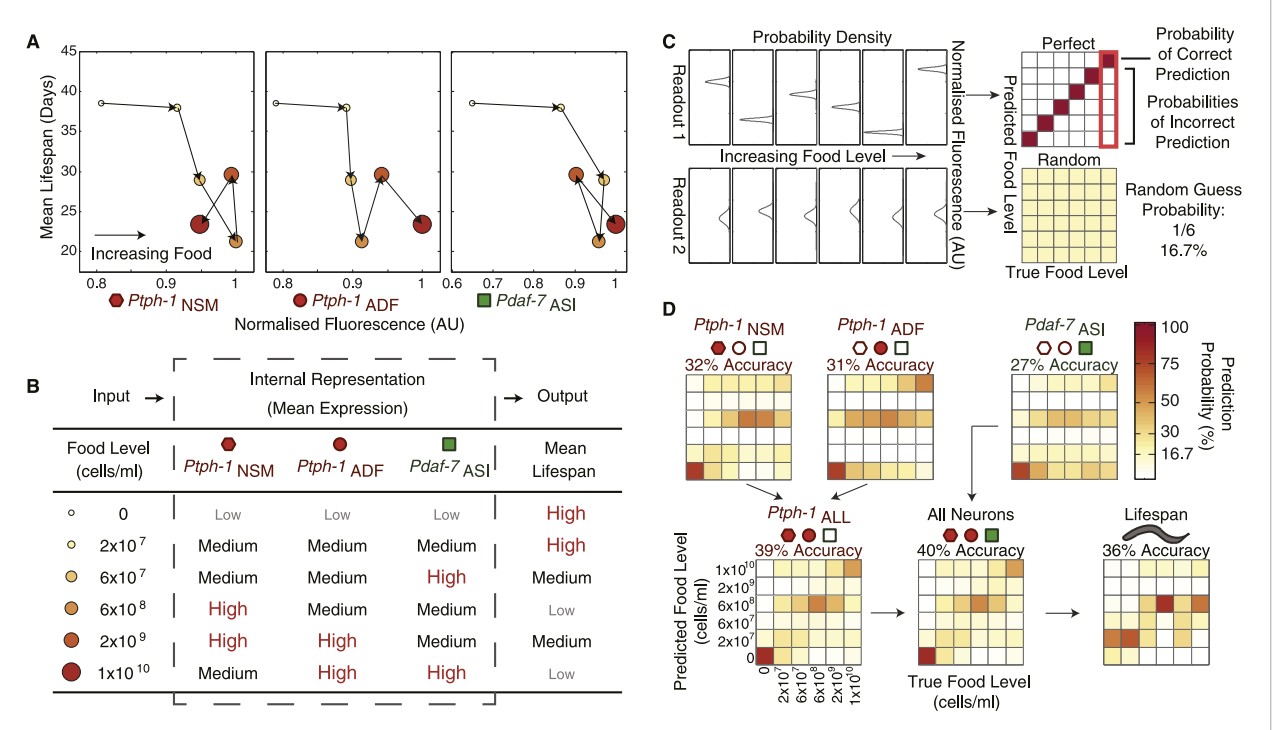

**Figure 4.** The combination of all neuronal gene-expression readouts generates a unique internal representation of food levels. (**A**) Relationship between individual gene-expression profiles and the lifespan responses across the six food levels indicates that the individual readouts are insufficient to uniquely specify lifespan responses. (**B**) The combination of *tph-1* in NSM (*Ptph-1*NSM) and ADF (*Ptph-1*ADF) and *daf-7* in ASI (*Pdaf-7*ASI) create a non-monotonic multivariate encoding scheme capable of both uniquely representing food inputs and potentially specifying lifespan outputs. (**C**) The ability of expression and lifespan readouts to respond to and represent (encode) food conditions can be estimated by using the readouts of interest to infer the true food conditions. The results can be visually represented by matrices where the squares in each column indicate the frequency with which particular inferences are made for a given true food level. Distinct, non-overlapping readouts result in high discriminatory power, represented by a highly diagonalized matrix (top). Indistinct, overlapping response profiles result in low (random) discriminatory power, represented by a uniform matrix (bottom). (**D**) Matrices indicating the representational capability of *tph-1* and *daf-7* readouts individually or in combination exhibit similar total encoding fidelity to that of lifespan outputs in wildtype animals.

First, *daf-7(−)* and *tph-1(−)* mutants had non-redundant phenotypes, indicating that they acted in parallel pathways (*Figure 5A*). For example, compared to the single mutants, the *tph-1(−)*; *daf-7(−)* double mutant showed a greater increase in the expression of *tph-1* in NSM at the higher food levels (*Figure 5A*, left). Second, we observed crosstalk between *tph-1* and *daf-7*. Loss of *daf-7* signalling affected *tph-1* expression in both NSM and ADF neurons (*Figure 5A*, left and middle) while loss of *tph-1* affected *daf-7* expression in the ASI neuron at higher bacterial concentrations (*Figure 5A*, right). Third, both *tph-1* and *daf-7* regulated their own expression. *tph-1* expression in NSM and ADF was altered in *tph-1(−)* mutants (*Figure 5A*, left and middle) and *daf-7* expression in ASI was altered in *daf-7(−)* mutants (*Figure 5A*, right). The self-regulation of *daf-7* and *tph-1* was not mediated via their crosstalk: *daf-7* influenced its own expression in the absence of *tph-1* activity (*Figure 5A*, right) and *tph-1* self-regulated in the absence of *daf-7* activity (*Figure 5A*, left and middle).

This intricate regulation had several profound effects on food encoding. First, the regulation between *daf-7* and *tph-1* shaped their responses to food. The relatively monotonic increase in *tph-1* expression in ADF with bacterial concentration in wildtype animals was specified by *daf-7* regulation; in *daf-7(−)* mutants, the response of *tph-1* in ADF became more non-monotonic (*Figure 5A*, middle). The non-monotonic response of *daf-7* expression in ASI was also altered by the loss of *daf-7* auto-regulation (*Figure 5A*, right). Since a non-monotonic encoding scheme provides theoretical advantages in representing the range of inputs (above), changes in the shape of these responses could impact how efficiently food is represented.

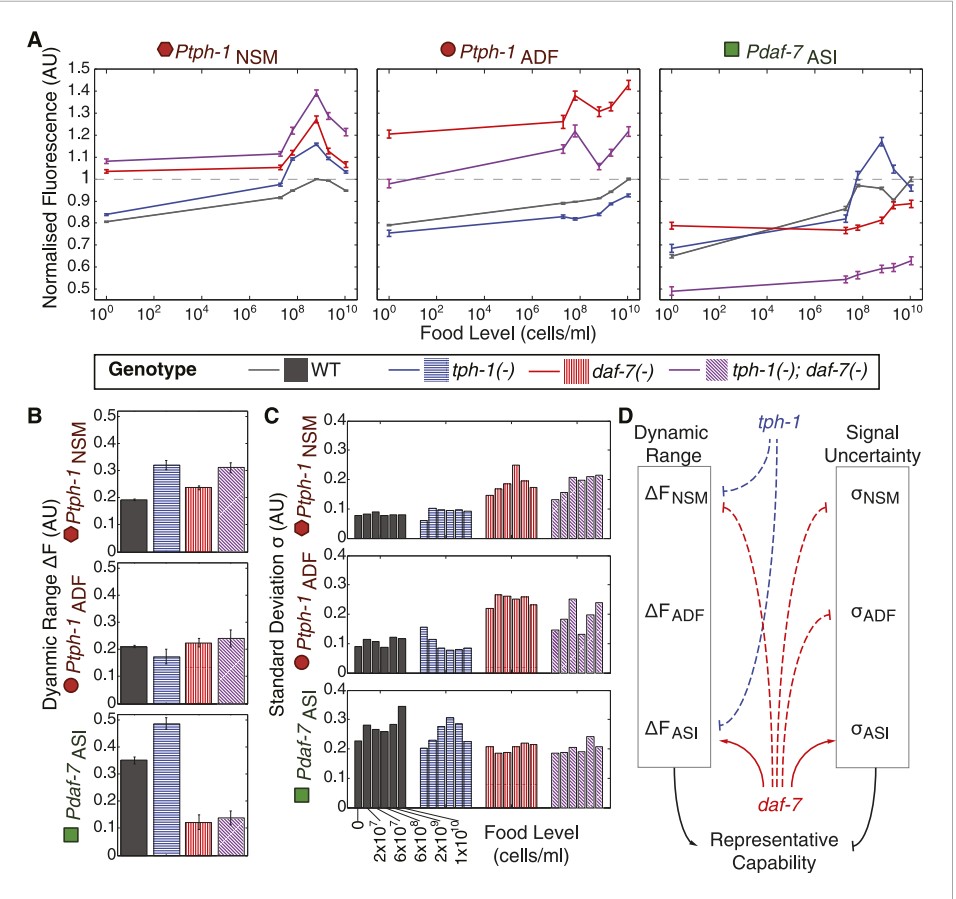

**Figure 5.** Crosstalk and self-regulation of *tph-1* and *daf-7* are important in shaping the pattern, range and variability of food-induced gene-expression responses. (**A**) Food-responsive expression profiles of *tph-1* in NSM (*Ptph-1*$_{NSM}$) and ADF (*Ptph-1*$_{ADF}$), and *daf-7* in in ASI (*Pdaf-7*$_{ASI}$) for different genetic backgrounds, as indicated in the legend. As in *Figure 3*, all values are normalized to the highest wildtype mean expression response observed in the respective neuron (dotted line). (**B**) Effects of *tph-1* and *daf-7* signalling on the dynamic range of food-induced expression modulation for *tph-1* in NSM and ADF and *daf-7* in ASI. The dynamic range (ΔF) is defined by the difference between the highest and lowest mean expression responses across the six food levels for each genotype. (**C**) Effects of *tph-1* and *daf-7* signalling on the inter-individual variability of expression responses as measured by the standard deviation of gene-expression distributions in each neuron. (**D**) Schematic of the total regulatory effects of *tph-1* and *daf-7* on both the dynamic range (ΔF) and variability and uncertainty (σ) of expression readouts to induce opposing effects on representational capability. *Figure 5—figure supplement 1* shows the distributions of gene expression for *Ptph-1*$_{NSM}$, *Ptph-1*$_{ADF}$, and *Pdaf-7*$_{ASI}$ across all genotypes tested.
The following figure supplement is available for figure 5:

**Figure supplement 1**. Neuron-specific expression distributions in wild type and mutants.

Second, these regulatory interactions also affected the dynamic range of the gene-expression profiles. Amplification of dynamic range may contribute to increased encoding capability by increasing separation of gene-expression signals. Compared to wildtype, mutations in *tph-1* and *daf-7* increased the dynamic range of *tph-1* expression in NSM but not in ADF, while the *daf-7(−)* mutation dramatically decreased the dynamic range of *daf-7* in ASI (*Figure 5B,D*).

Third, besides affecting average gene expression, the *daf-7(−)* mutation also had a strong effect on the inter-individual variance of each gene-expression distribution (*Figure 5C,D* and *Figure 5—figure supplement 1*). In contrast to the effects of dynamic range, increasing inter-individual variation degrades encoding capability by increasing overlap between gene-expression signals. At several food levels, *daf-7(−)* mutants increased the variance of *tph-1* expression compared to the wildtype or the

*tph-1(−)* single mutant, whilst the *daf-7(−)* mutation reduced the variance of *daf-7* expression in ASI (*Figure 5C*). This result provided an interesting example of regulated inter-individual variability in gene expression that is associated with a physiological output in a metazoan. Thus, the shape, dynamic range and inter-individual variance of the gene-expression responses could all be regulated by interactions between *daf-7* and *tph-1*.

## Regulatory interactions between *daf-7* and *tph-1* tune encoding fidelity

To understand the multifaceted effects of *daf-7* and *tph-1* signalling on fidelity, we computed the encoding accuracy of neuronal gene expression in *daf-7(−)* and *tph-1(−)* mutants using the same maximum likelihood framework as applied to the wildtype animals (*Figure 4C,D*, and *Figure 6—figure supplement 1*). *tph-1(−)* mutants showed increased encoding capability in all readouts while *daf-7(−)* mutants demonstrated consistently diminished fidelity (*Figure 6A*), consistent with their respective effects on dynamic range and the variability of gene expression (*Figure 5B,D*). The intermediate encoding capability of the double mutants reflects the additive effects of the mutations with regard to encoding (*Figures 5B,C, 6A*). As with wildtype animals, combining information from multiple neurons tended to increase the accuracy in these mutants and *tph-1* expression in the NSM and ADF neurons accounted for the majority of this representational capability (*Figure 6B*). These results showed that *tph-1* and *daf-7* served mechanistically distinct and opposing roles in modulating the fidelity of this food representation (*Figure 6C,D*). Whilst the transcriptional responses of *daf-7* itself added little encoding capability overall (*Figures 4D, 6B*), its regulatory role in reducing gene-expression variability facilitated the encoding power of *tph-1* (*Figure 6B,C*). Conversely, *tph-1* outputs served a major encoding role in the system (*Figure 6B*), but its regulatory effect decreased the dynamic range of transcriptional responses (*Figure 5B*) and limited the encoding accuracy of the system in wildtype animals (*Figure 6C*). Together, our results reveal the roles of cross- and self-regulation within this circuit, and the mechanisms by which *tph-1* and *daf-7* tune performance.

While our reporter system allowed us to ascertain all transcriptional outputs in all genetic backgrounds, not all of these readouts resulted in functional outputs in the mutants. For example, in *daf-7(−)* mutants, *daf-7* expression in ASI did not yield a functional protein that could mediate physiological outputs. When we accounted for the loss of the corresponding functional genetic outputs in the mutants, the overall functional representational capability was consistently diminished in comparison to wildtype (*Figure 6E*). The loss of representational capability in either *tph-1(−)* or *daf-7(−)* mutants confirms that these genes have non-redundant encoding functions via two distinct mechanisms. *tph-1(−)* mutants showed diminished functional representation primarily due to loss of *tph-1* as a functional output whereas *daf-7(−)* mutants showed diminished functional representation primarily due to increased variability in *tph-1* expression (*Figure 6A,B,E*).

Remarkably, we found corresponding decreases in the accuracy of both the functional representations and lifespan outputs when *daf-7* and *tph-1* were mutated (compare *Figure 6E,F*). This result is consistent with the disruption in internal representation and the reduction in lifespan-responsiveness to food in the mutants (*Figure 1C,E*). We note that *tph-1(−)*; *daf-7(−)* double mutants retained some response to food levels (*Figures 1C, 6F*) indicating the existence of additional unknown gene(s) that encode food. Nonetheless, the significant and correlated effects of mutating *daf-7* and *tph-1* on the accuracy of both gene-expression and lifespan readouts (*Figure 6E,F*) strongly indicate that *tph-1* and *daf-7* act as a substantial part of a multi-neuron system linking environment to lifespan.

## Discussion

We have shown that food abundance is functionally encoded in vivo in *C. elegans* by the combinatorial expression of *tph-1* and *daf-7* in three neuron pairs. Lifespan increases and decreases as a function of food abundance. The phenotypes of *tph-1(−)* and *daf-7(−)* mutants reveal that rather than serving intrinsic roles in lifespan extension, serotonin and TGFβ are signals required for bidirectional food-dependent lifespan changes (*Figure 1D*). This bidirectional phenotype was only revealed by analysis of lifespan across a very broad range of food levels. Many genes have been implicated in lifespan modulation or DR-related responses; a systematic analysis under additional conditions may provide new insights on their roles in food-dependent responses. That *daf-7* and *tph-1* act in food-responsive sensory neurons, together with the bidirectional attenuation of lifespan responses in the single and double mutants suggests that loss of *tph-1* and *daf-7* impairs the ability to sense and/or convey

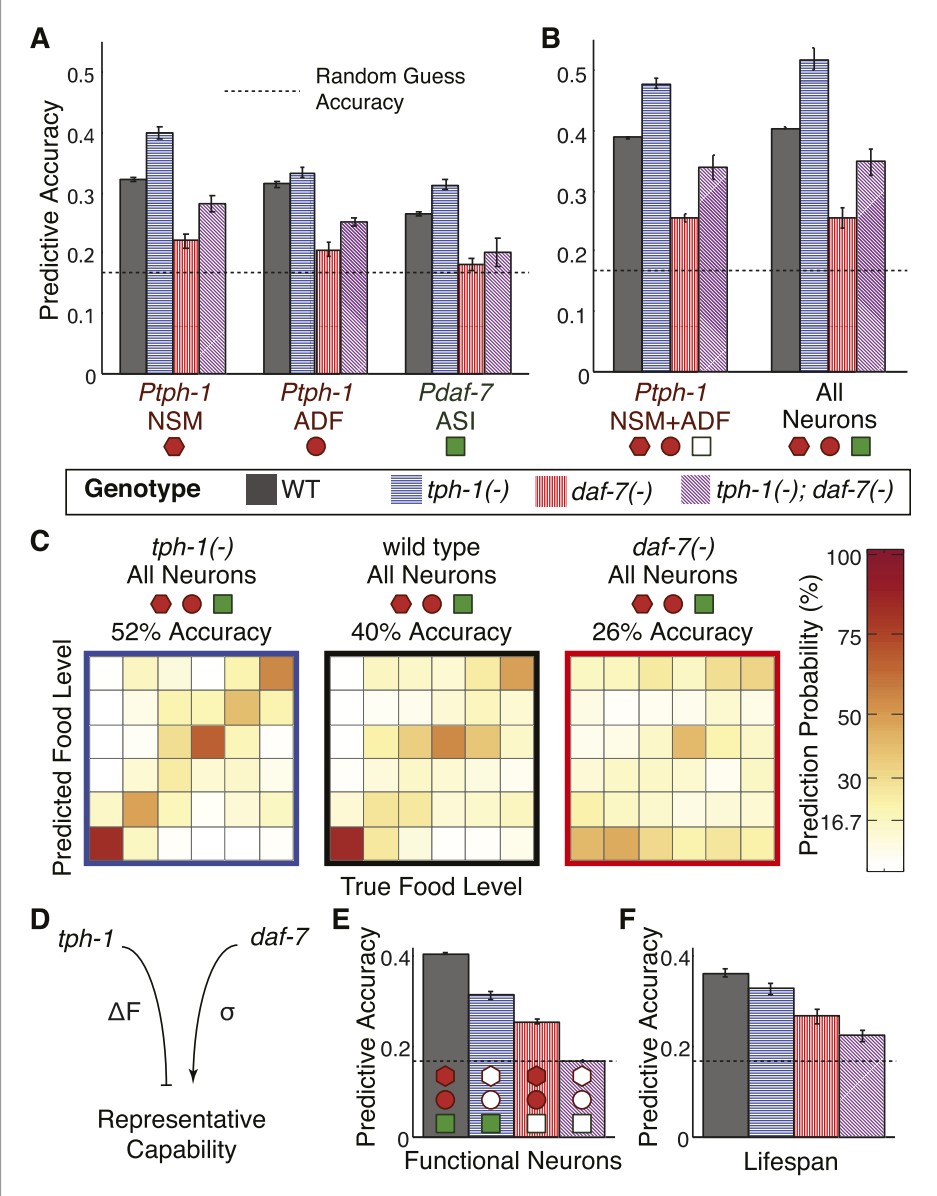

**Figure 6.** Cross- and self-regulation of *tph-1* and *daf-7* control the accuracy of internal representation of food levels. (**A**) Encoding accuracy of individual neuron-specific expression readouts in wildtype and mutant populations, as indicated by the legend. Dotted line indicates the lower bound for encoding accuracy due to chance. (**B**) Functional combinations of the neuron-specific expression readouts increase encoding accuracy in both wildtype and mutant populations. (**C**) Matrices indicating the full encoding accuracy of the combination of all gene-expression readouts in wildtype and mutant animals reveal a surprising increase in accuracy with the loss of *tph-1*. (**D**) Schematic indicating the distinct mechanisms by which *tph-1* and *daf-7* control the representational capabilities of the system. *tph-1* and *daf-7* exert their effects largely via modulating dynamic range or variability in gene expression, respectively. (**E**) Mutant animals show diminished encoding accuracy relative to wildtype when only functional expression readouts (filled symbols) are considered. For example, only *daf-7* expression in ASI (*Pdaf-7*<sub>ASI</sub>) is a functional readout in the *tph-1*(–) mutant. (**F**) Encoding accuracy of lifespan responses in the mutants exhibit decreases that are consistent with the loss of representational capability in (**E**). *Figure 6—figure supplement 1* shows the decoding analysis for all neuron combinations across all genotypes using both maximum likelihood-based and probability-based decoders.

The following figure supplement is available for figure 6:

**Figure supplement 1.** Food encoding accuracy of all neuron combinations and lifespan outputs for wildtype and mutant animals.

information about food levels, leading to a flattened response (*Figure 1C*). This interpretation is further supported by the observations that loss of *daf-7* and *tph-1* reduced the dynamic range of lifespan responses (*Figure 1E*), and lowered the accuracy of food level representation (*Figure 6E,F*). *tph-1* and *daf-7* have established functional connections to conserved pathways for regulating metabolism, cellular maintenance, reproduction and longevity, such as the insulin/IGF pathway (*Liang et al., 2006*; *Shaw et al., 2007*; *Narasimhan et al., 2011*). Thus, this internal perception of food availability may facilitate the coordination of energy utilization and appropriate physiological and behavioral responses to optimize survival in variable environments.

How much information is conveyed by *daf-7* and *tph-1*? Several results indicate that *daf-7* and *tph-1* account for a substantial if not the majority of the food response in lifespan modulation. First, loss of both *tph-1* and *daf-7* reduced the dynamic range of the lifespan response to food by ∼60% (*Figure 1E*). Second, the decoding analysis, which systematically considers the overlap between multiple response distributions (*Figure 4*), showed that much of the information about food levels is lost in the *tph-1(−); daf-7(−)* double mutant: the predictive accuracy of the lifespan response drops to 5% above random (*Figure 6—figure supplement 1H*). Third, the predictive accuracy of gene expression and lifespan are heterogeneous—certain food levels were correctly predicted with far higher accuracy (>75%) than the average accuracy (see decoding matrices in *Figures 4D, 6C* and *Figure 6—figure supplement 1*). This heterogeneous performance is due in part to non-monotonic expression responses (*Figure 3C*), as some food levels lead to similar responses. Fourth, it is important to note that an encoding system only needs to match the accuracy of the output to be good enough for regulating a biological process. The mean accuracy of lifespan responses to food thus provides a biologically relevant limit on the required performance for encoding food levels. In this light, the comparable mean accuracy between *daf-7*/*tph-1* gene expression and lifespan indicates that the performance of the coding system is well suited to the accuracy of the process it modulates. Fifth, we found corresponding drops in the predictive accuracy of both gene expression and lifespan across *daf-7(−)* and *tph-1(−)* mutants (compare *Figure 6E,F*). This result is particularly remarkable because the predictive accuracy depends on not just the means but also variances. While these results support the idea that *daf-7* and *tph-1* functionally encode significant information about food levels, the *tph-1(−); daf-7(−)* double mutant still retains some information about food, as shown by its lifespan response to food and its predictive accuracy of food (*Figures 1C, 6F*). Therefore additional molecule(s) exist that cooperate with *daf-7* and *tph-1* in food encoding. These may include previously characterized DR regulators or undiscovered genes.

Our findings provide a more nuanced view of the interactions between the serotoninergic and TGFβ neuroendocrine systems. Genetic epistasis between *tph-1* and *daf-7* had suggested that these two genes functioned in the same pathway (*Sze et al., 2000*; *Chang et al., 2006*). Our results indicate that *tph-1* and *daf-7* are likely to act in parallel to modulate lifespan in response to food, and that their mutual regulation represents crosstalk between these two pathways. First, we found that their epistatic relationships are food-dependent rather than hard-wired. This result is reminiscent of another food-dependent epistasis between neuropeptide and catecholamine receptors (*npr-1* and *tyra-3* respectively) that act in *C. elegans* sensory neurons (*Bendesky et al., 2011*). Perhaps crosstalk during sensory processing leads to more complex genetic interactions. Second, *tph-1* and *daf-7* play distinct regulatory roles in shaping the food representation and tuning discriminatory power (*Figures 5D, 6D*; discussed below). Third, *daf-7* and *tph-1* show additive phenotypes in both gene-expression responses and predictive accuracy of lifespan (*Figures 5A, 6F*). Since the predictive accuracy reflects the information content on food, the additive loss indicates that *daf-7* and *tph-1* carry non-redundant information about food abundance. Together, these results point to parallel rather than sequential signalling for serotonin and TGFβ.

Conceptually, the genetic regulation between serotonin and TGFβ is similar to feed-forward and feedback regulatory architectures that occur in other neural circuits operating on a shorter time-scale (*Harris and Mrsic-Flogel, 2013*; *Wilson, 2013*), and analogous mechanisms to regulate information transduction may be used for different sensory modalities. Different regulatory interactions between *daf-7* and *tph-1* modulate encoding capacity in specific ways, revealing novel system-level roles for these interactions. *tph-1* activity limits the dynamic range, and consequently the representative capacity of *tph-1* and *daf-7* expression in response to food. It is possible that the increased dynamic range of *daf-7* expression in *tph-1(−)* animals may partially compensate for the loss of *tph-1* as a major functional encoder. Surprisingly, *tph-1* self-regulation indicates that *tph-1* and serotonin

signaling is likely to self-limit representative capacity by reducing the dynamic range. In contrast, the cross-regulatory effect of *daf-7* is necessary to reduce variability in *tph-1* expression, which represents a new component of *tph-1* expression that is regulated by *daf-7*. The parity in encoding performance between *tph-1* and *daf-7* expression and lifespan (*Figure 6E,F*), along with the attenuation of lifespan response in the mutants, suggests that this neuronal representation has major functional information-carrying capacity. Distributing a combinatorial code across multiple neurons with distinct responses ensures robustness of signal fidelity against variability in each neuron-type, analogous to population codes in certain sensory circuits (*Singer and Gray, 1995*). Furthermore, a distributed code enables multicellular organisms to implement additional layers of regulation unavailable to unicellular organisms.

Inter-individual variability in gene expression has been extensively studied in single cell systems where it plays key roles in diverse processes (*Eldar and Elowitz, 2010*). By measuring this variation in neurons using high-throughput quantitative microscopy, we highlight implications of this variability in encoding food levels in nervous system, which have not been extensively explored in intact animals and are likely to be generalizable to other neural processes. First, gene-expression variability limits the accuracy of neural food representations. Second, this variability can be regulated, as shown by the role of TGFβ in reducing variability in serotonin signaling to increase fidelity. Third, in this TGFβ and serotonin circuit, the effects of variability are tempered by the dynamic range of responses (which can reduce the overlap between broad distributions) and by the use of multiple neurons. Thus, the overall fidelity of information processing is shaped at least in part by an interplay of variability and other parameters.

Neural encoding has traditionally been studied in the context of short-term electrical activity. However, activity-dependent transcription has been studied in diverse forms of neural plasticity that impact behavior, learning and memory (*Zhang et al., 2005*; *Chang et al., 2006*; *Flavell and Greenberg, 2008*; *Pocock and Hobert, 2010*). Our results suggest that gene expression in a multi-neuron system can also encode food abundance to tune physiological states at longer time scales. In general, the complexity of gene-expression responses, such as those observed in the songbird forebrain during song communication, suggest that such responses can perform computations (*Clayton, 2013*). It is likely that additional layers of computation in the nervous system, such as the gene-expression encoding shown here, are necessary to mediate longer-term behavior and physiology.

## Materials and methods

### Strain construction and transgenesis

The following parental strains were used to generate all other used in this study: N2 (wildtype), QL101 *tph-1(n4622)* II, QL282 *daf-7(ok3125)* III, EG6701 *ttTi4348* I; *unc-119(ed3)* III; *oxEx1580*, EG6699 *ttTi5605* II; *unc-119(ed3)* III; *oxEx1578*. Worms were cultured according to standard protocols (*Stiernagle, 2006*).

The reporters for *daf-7* and *tph-1* expression were generated from fosmid clones carrying the wildtype *tph-1* and *daf-7* genomic sequences (Source BioScience Lifesciences, UK). We used a recombineering pipeline (*Sarov et al., 2006*) to replace the coding sequence of the target gene in the fosmid with either mCherry or Venus. We then subcloned the reporter constructs flanked by the native 5′ and 3′ intergenic regions of *daf-7* or *tph-1* respectively into pCFJ352 and pCFJ151 for integration into defined positions of the *C. elegans* genome using MosSCI (*Frokjaer-Jensen et al., 2008*). The *tph-1::mCherry* reporter (*drcSi61*) was integrated on chromosome I at the *ttTi4348* Mos locus. The *daf-7::Venus* reporter (*drcSi7*) was integrated on chromosome II at the *ttTi5605* Mos locus. We used standard genetic techniques to construct strains carrying both fluorescent reporters in various genetic backgrounds.

Cell-specific rescue lines used in *Figure 3* were generated in two ways. The *tph-1* ADF- and NSM-expression constructs were from *Zhang et al. (2005)*. The ASI-specific *daf-7* construct was generated in two steps; first we constructed a GFP reporter driven by the promoter of *gpa-4*, which is specifically expressed in ASI (*Jansen et al., 1999*), using the recombineering pipeline (*Sarov et al., 2006*) above. This was then subcloned into the pCFJ151 vector for MosSCI targeting of *ttTi5605* locus (*Frokjaer-Jensen et al., 2008*). This recombinant plasmid was then digested with NheI and AvrII to linearize it for use as a vector backbone. Using a *daf-7*-containing fosmid as template, we amplified a PCR

product that contained the entire *daf-7* gene from start to stop codon including its native introns and added overlapping linkers to the *gpa-4* 5′ and 3′ UTRs at the respective ends of the PCR product (Primers DP101, 5-<u>TCAACAAGCTCAGGAGGTAGCGGC</u>GATGTTCATGGCATCTTCAC-3 and DP102 5-<u>TCTAGTTAAACGT</u>TTATGAGCAACCGCATTTC-3, underlined sequences represent *gpa-4* linkers). A second PCR product was then amplified from the undigested recombinant *gpa-4*::GFP reporter plasmid above, which contained the entire *gpa-4* 3′ intergenic sequence and added linkers to the *daf-7* gene and the multiple cloning site (MCS) of the pCFJ151 vector (Primers DP103 5-<u>CGGTTGCTCATAA</u>ACGTTTAACTAGATTAGCACAAATG-3 and DP104 5-<u>TTGACTAGAGGGTAC CAGAGCTCAC</u>TAAATAATAAAAAACCTAAAATTGTTTC-3, underlined sequences represent *daf-7* and MCS linkers, respectively). These fragments were then seamlessly joined into the digested vector backbone using Gibson assembly (New England Biolabs, Ipswich, MA) to yield the final vector, which contained the *daf-7* gene flanked by all the regulatory elements of *gpa-4*. To generate transgenic animals carrying these cell-specific rescue constructs we injected them into both N2 and *tph-1(n4622)*; *daf-7(ok3125)* double mutants at a concentration of 10 ng/µl along with an *ofm-1*::GFP reporter (*Miyabayashi et al., 1999*) at 25 ng/µl. Transgenic animals were maintained by selecting for GFP-positive animals under a fluorescent dissecting scope.

## Reporter validation

Our *daf-7* and *tph-1* reporters contain similar regulatory sequences, and are expressed in the same patterns, as reporters that have been extensively validated in the literature. *tph-1* reporters have been validated by experiments showing corresponding changes under a variety of environmental conditions, including starvation (*Liang et al., 2006*; *Cunningham et al., 2012*), exposure to pathogenic bacteria (*Zhang et al., 2005*), and hypoxia (*Pocock and Hobert, 2010*). The supplemental information in *Liang et al. (2006)* showed that serotonin levels in ADF were reduced in animals starved for 8 hr. We further verified that serotonin levels in NSM of wild type adults were also reduced after food deprivation, in agreement with the behavior of the P*tph-1*::mCherry reporter. The serotonin levels were determined by anti-serotonin immunofluorescence staining in 4 day old adult worms. Worms subjected to our DR protocol at two food levels (no food and $6 \times 10^8$ cells/ml) were used for immunostaining using an anti-serotonin primary antibody (Immunostar Inc., Hudson, WI) and Alexa594 coupled secondary antibody as previously described (*Loer and Kenyon, 1993*). Day 4 adults deprived of bacteria showed less serotonin staining ($1.19 \pm 0.29$ AU; n = 26) than those exposed to $6 \times 10^8$ cells/ml of bacteria ($1.94 \pm 0.28$ AU; n = 21); these differences were significant ($p = 1.38 \times 10^{-11}$; Students T-Test). *daf-7* reporters have been previously validated by experiments showing corresponding changes in both *daf-7*::GFP and *daf-7* mRNA when switching from starved to fed states, under pheromone induction of dauer, and during exit of dauer (*Ren et al., 1996*). In particular, *daf-7* is down-regulated upon starvation, and both *daf-7*::GFP and *daf-7* mRNA levels increase when starved animals are fed (*Ren et al., 1996*), consistent with our observations (*Figure 3*).

For expression studies in middle-aged adults used in this study, we found that using fluorescent reporters was the most reliable method. *daf-7* and *tph-1* are expressed in a small number of cells, resulting in very low mRNA levels that were not reliably measured by real-time quantitative PCR because the signals were close to background (data not shown). Furthermore, procedures for whole-mount immunohistochemistry (*Loer and Kenyon, 1993*) and single-molecule fluorescence in situ hybridization (*Raj et al., 2008*) resulted in high background fluorescence that precluded accurate measurements in older adults using these methods.

## Bacterial cultures for DR

Our DR protocol was derived from two previously published methods for DR on agar plates (*Greer et al., 2007*; *Ching and Hsu, 2011*). After growing of *E. coli* (OP50) overnight in LB at 37°C shaking, the bacterial cultures were shocked with streptomycin at 50 µg/ml for 30 min whilst shaking and then chilled on ice for 15 min before being centrifuged at 4500×*g* for 25 min. The supernatant was then decanted and the bacteria were resuspended in S Basal containing streptomycin (50 µg/ml). The volume of S Basal + streptomycin required for resuspension was determined by measuring the OD$_{600}$ of a 10-fold dilution of the overnight culture prior to centrifugation and then calculating the volume required to give the resuspended culture a theoretical OD$_{600}$ of 56, which corresponds to a total cell count (live + dead) of $\sim 1 \times 10^{10}$ cells/ml. Serial dilutions of this high-density stock were used to derive all subsequent bacterial concentrations. Stocks were stored at 4°C until plates were ready to seed.

We used NGM plates supplemented with both streptomycin and carbenicillin, each at 50 μg/ml. The use of dual antibiotics ensured that bacteria were unable to proliferate on the agar plates. Plates were seeded with a dispensing pipette to ensure that all plates received an equal volume of liquid. Plates were typically seeded 2–3 days in advance to allow the bacterial lawns to dry out before worms were put on the plates.

## C. elegans culture for lifespans

Many *C. elegans* ageing studies use fluoro-2′-deoxyuridine (FuDR) to inhibit the germline of experimental animals in order to eliminate progeny that may otherwise confound analyses. However, the use of FuDR is problematic as the germline in *C. elegans* is a major regulator of longevity (*Hsin and Kenyon, 1999*; *Lin et al., 2001*) and its use can cause gene-specific effects on lifespan (*Aitlhadj and Sturzenbaum, 2010*). Our protocol uses a novel method to eliminate production of progeny by inhibiting formation of the eggshell of fertilized *C. elegans* embryos through RNAi of *egg-5* (*Cheng et al., 2009*; *Parry et al., 2009*) resulting in their death. Several results indicate that *egg-5(RNAi)* does not affect food-dependent lifespan responses. We observed similar lifespan responses for wildtype animals in experiments where *egg-5(RNAi)* was omitted (*Figure 1—figure supplement 1*). Moreover, we also observed similar lifespan responses under dietary deprivation (*Kaeberlein et al., 2006*; *Lee et al., 2006*) and at the intermediate to high food levels where lifespans have been measured in other publications (*Greer and Brunet, 2009*; *Greer et al., 2009*).

Lifespan assays were performed on 6 cm CellStar (Greiner Bio-One, UK) plates at density of 15 worms that were passaged to fresh plates by manual transfer using a platinum wire pick. To avoid physical damage to the worms during transfer, animals were floated off the pick by immersing it into a 10 μl droplet of S Basal + streptomycin on the surface of the new plate. NGM agar plates containing carbenicillin and streptomycin were always seeded with a single 225 μl aliquot of the appropriate bacterial concentration resuspended in S Basal + streptomycin, while plates with 0 cells/ml, equivalent to dietary deprivation, were seeded with S Basal + streptomycin as a control.

Animals for all strains were raised on live OP50 bacteria for two generations at 20°C. Synchronized L4-stage progeny of the F2 parents were manually transferred to NGM plates supplemented with 1 mM IPTG and 50 μg/ml carbenicillin that were seeded with HT115 bacteria expressing dsRNA targeting *egg-5*. Animals were exposed to *egg-5* RNAi for 24 hr before being transferred to NGM + streptomycin + carbenicillin plates seeded with our baseline food concentration of $2 \times 10^9$ cells/ml. On day 2 of adulthood, animals were shifted to the desired DR food level and temperature and then transferred at regular intervals to fresh plates until day 14 of adulthood according to a defined scheme (*Figure 1—figure supplement 2*). We initiated DR on day 2 adults because it had an intermediate effect on lifespan as compared to initiation on day 1, 3, and 4 (*Figure 1—figure supplement 1*). We reasoned this intermediate effect served as a potentially more sensitive window to detect factors that modify food responses, as the effects of food were non-saturating (unlike the effect of day 3 DR initiation, which was indistinguishable from day 4 DR initiation). Animals were scored for movement upon gentle prodding with a wire pick; failure of response was scored as death. Animals were scored for death at every transfer point and then daily after the last transfer point. Lifespans were subjected to Kaplan–Meier analysis and significance was assessed using both Log-Rank and Wilcoxon tests.

## C. elegans culture for imaging

Animals in imaging experiments were cultured on 10 cm CellStar (Greiner Bio-One) to allow for a higher number of worms (∼100) than lifespan assays. For quantitative imaging, animals were grown and subjected to DR under identical conditions to those assayed for lifespan, except that animals were washed from plate to plate with the same schedule instead of manual transfers. Due to the greater number of worms in these assays, animals were transferred between plates by washing with S Basal + streptomycin. 10 cm plates were seeded with five 225 μl aliquots of bacteria or S Basal + streptomycin in a cross-like formation.

In our initial imaging experiments using the *daf-7* and *tph-1* reporters in a wildtype background, L1-stage larvae were synchronized collecting animals that hatched in a 2-hr window and transferring them to fresh 10 cm NGM plates seeded with OP50 at 20°C. These animals were then harvested 36 hr later once they had reached the L4 stage and washed on to plates for *egg-5* RNAi. After 24 hr on the RNAi plates the animals were washed to the baseline food level of $2 \times 10^9$ cells/ml for 1 day before the

initiation of DR on the second day of adulthood. Animals were transferred to fresh plates on the third and fifth days of adulthood and imaged on the sixth day.

Imaging experiments involving comparisons to strains containing the *daf-7(ok3125)* mutation could not be treated as above, as this mutation causes a severe egg-laying defect. Instead, strains for these experiments were grown for 2 generations as before, and then the gravid F2 adults were collected and treated with Sodium Hypochlorite to break open the animals and liberate their eggs (*Stiernagle, 2006*). The eggs were then deposited onto NGM plates seeded with OP50 for either 72 hr, in the case of strains in a *daf-7(ok3125)*-containing background, or 48 hr for non-*daf-7(ok3125)*-containing strains. L4-stage animals were then harvested after these respective intervals and then treated exactly as above.

## Microfluidic imaging

For our quantitative imaging studies, animals subjected to our DR protocol were imaged at day 6 of adulthood using a custom microfluidic platform (*Chung et al., 2008*; *Crane et al., 2012*). Animals were suspended in S Basal + streptomycin on day 6 of adulthood and introduced into our custom microfluidic device via pressure driven flow (*Chung et al., 2008*; *Crane et al., 2012*). Briefly, microfluidic devices were manufactured in polydimethylsiloxane using standard multilayer soft lithographic techniques (*Unger et al., 2000*) and covalently bonded to a glass coverslip via oxygen plasma treatment. On the device, individual animals were sequentially directed into and trapped within an imaging channel gated by pressure-driven on-chip valves (*Unger et al., 2000*) under the control of custom LabVIEW software (*Figure 3—figure supplement 1*). Dense, 2-micron fluorescent z-stacks through head of each worm were collected using a standard epifluorescence microscope (Nikon Ti-E inverted microscope) with a 40× oil objective (1.3 NA) and a Hamamatsu Orca R2 camera. Red and green fluorescent intensities for each of our fluorescent reporters were collected simultaneously using an Optosplit II emission splitter and stored for analysis. We automated the image acquisition and image processing using custom LabVIEW and MATLAB scripts; the analysis software is deposited at https://github.com/meizhan/SVMelegans.

## Image processing

Z stacks from our quantitative imaging studies were loaded into MATLAB to be analyzed for single-cell expression (*Figure 3—figure supplement 1*). To identify neuron-pairs and their locations within the imaging plane, maximum projections were computed and a thresholding algorithm was utilized to locate individual fluorescent cells. Identifications of the cells were then computed based on relative distances and locations within the worm head. For quantification, the three dimensional volume around each cellular location was extracted from the full z-stack and intensity was integrated over a consistent number of the brightest pixels, which fully encapsulate the entire cell in all cases. To avoid potential interference from condition-specific changes in the gut auto-fluorescence, the background intensity was calculated for cell pairs near the gut (ADF, ASI), via estimation of the mode of the intensity distribution in a volume around the neuron. This background intensity value was subtracted from the integrated fluorescence to obtain the final output.

## Decoding analysis

To estimate the ability of our gene-expression readouts and lifespan responses to encode information about the food inputs, we applied a Bayes classifier with fivefold cross validation (*Dayan and Abbott, 2005*). To do this we randomly segmented the data into five test groups. For each test group, we computed expression or lifespan probability distributions for each food level based on the remaining data (the training set). We employed a multivariate Gaussian distribution to fit the expression data and a Weibull distribution for lifespan data. Using these probability distributions, we then calculated the conditional probability that each member of the test group had their particular expression or lifespan output if they had been subjected to each of the food levels. The accuracy of these raw probability values can be found in *Figure 6—figure supplement 1* panels A, C, E and G. To make a final inference about the food level each worm was subjected to, the animal was assumed to have come from the food level with the highest conditional probability. The encoding accuracy of this maximum likelihood model can be found in *Figure 6—figure supplement 1* panels B, D, F, and H. As with all measurements of variance, these accuracy estimates should be considered as lower bounds due to experimental noise.

In the decoding analysis for expression data, we limited the data from *Figure 5—figure supplement 1* to animals with corresponding data from all three neuron pairs ($N \geq 726$ for WT, $N \geq 110$ for *tph-1(–)*, $N \geq 83$ for *daf-7(–)* and $N \geq 54$ for *tph-1(–);daf-7(–)* for each food level). For the lifespan data, we used the pooled aggregate of all of our lifespan data (highlighted rows in *Figure 1—source data 1* for wildtype and pooled mutant data in *Figure 1—source data 2*).

## Acknowledgements

We are grateful for reagents from the Horvitz Lab, the Bargmann Lab, and the *C. elegans Genetics Center*, which is funded by NIH Office of Research Infrastructure Programs (P40 OD010440). We thank M Dionne, T Streelman, P McGrath, M Styczynski, G Stanley, Y Zhang, C Bargmann, I Chou and the members of our labs for critical comments on this manuscript. We also thank N Lesica for suggesting computational neurobiology approaches for data analysis and comments on this manuscript; B Hanley, K Gers-Barlag and H Tunbak for assistance in generating some of the reagents and/or experiments; and B Parker and J Andrews for custom machining critical pieces of instrumentation. This research was supported by the Wellcome Trust (Project Grant 087146 to QC), BBSRC (BB/H020500/1 to QC), European Research Council (NeuroAge 242666 to QC), US National Institutes of Health (R01AG035317 and R01GM088333 to HL), and US National Science Foundation (0954578 to HL, 0946809 GRFP to MZ).

## Additional information

### Funding

| Funder | Grant reference | Author |
|---|---|---|
| Wellcome Trust | Project Grant 087146 | QueeLim Ch'ng |
| Biotechnology and Biological Sciences Research Council (BBSRC) | BB/H020500/1 | QueeLim Ch'ng |
| European Research Council | NeuroAge 242666 | QueeLim Ch'ng |
| National Institutes of Health | R01GM088333 | Hang Lu |
| National Science Foundation | 0954578 | Hang Lu |
| National Institutes of Health | R01AG035317 | Hang Lu |
| National Science Foundation | 0946809 GRFP | Mei Zhan |
| National Institutes of Health | Office of Research Infrastructure Programs (P40 OD010440) | Hang Lu |

The funders had no role in study design, data collection and interpretation, or the decision to submit the work for publication.

### Author contributions

EVE, DSP, MZ, Conception and design, Acquisition of data, Analysis and interpretation of data, Drafting or revising the article, Contributed unpublished essential data or reagents; AJS, Analysis and interpretation of data, Drafting or revising the article; HL, QC, Conception and design, Analysis and interpretation of data, Drafting or revising the article

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
