## [Decision Letter]

Thank you for sending your work entitled “A Gene-Expression-Based Neural Code for Food Abundance that Modulates Lifespan” for consideration at *eLife*. Your article has been favorably evaluated by a Senior editor, Oliver Hobert (Reviewing editor), and three reviewers.

The Reviewing editor and the reviewers discussed their comments before we reached this decision, and the Reviewing editor has assembled the following comments to help you prepare a revised submission.

Your manuscript characterized the correlation between the abundance of environmental food availability, lifespan and neuronal expression levels of tryptophan hydroxylase gene *tph-1* and TGF-beta *daf-7* in *C. elegans*. Neuronal influence on longevity is an important research area, thus this work is of significance. Although the effects of food signals on *tph-1::gfp* and *daf-7::gfp* expression and 5-HT and *DAF-7* on ageing have been studied before, this work provides several novel insights, by generating single copy *tph-1::gfp* and *daf-7::gfp* transgenic lines and systematically monitoring the expression levels in animals grown on a wide-range of *E. coli* food concentrations. The elegant experimental approach and detailed analysis are the strength of this paper.

However, there are a number of concerns about data interpretation that we ask you to fix in a revised version of the manuscript. In no order of importance:

1) One concern pertain to what seem to be modest phenotypic effects of *tph-1* and *daf-7* mutations on life span in the presence of food (with preservation of the bidirectional influence of varying food concentrations on life span), a relatively low predictive accuracy of reporter gene expression, and the low magnitude of changes in reporter gene expression observed (both in wild-type animals over the range of food concentrations tested and in *tph-1* and *daf-7* mutant animals compared to wild-type animals). These observations lead me to wonder if the authors are focusing on molecules that play an ancillary role in representing food abundance, to the exclusion of undiscovered molecules that are more central to this process. This should be discussed.

2) The authors repeatedly argued that 5-HT and *DAF-7* act in parallel to modulate the lifespan. However, a double mutation of *tph-1* and *daf-7* did not produce an additive ageing phenotype (Figure 1). Please discuss.

3) The authors made great efforts to establish differential expression *of tph-1::gfp* in ADF and NSM, particularly when worms grown on very high concentrations of *E. coli* food (Figures 3, 4 and 5). While the results are very interesting, the differential *tph-1* expression does not appear to play a central role in ageing. First, transgenic expression of *tph-1* in ADF or NSM restored the lifespan as a function of food availability. Second, there is no indication that the expression levels of those transgenes are modulated by food availability. Thus food restriction-induced changes in *tph-1* expression probably influence other 5-HT signaling pathways but not the ageing processes.

4) Why do *daf-7* and *tph-1* mutants cause extension of lifespan at some food concentrations but reduce it at others? Please discuss.

5) The sole reliance on gene expression, while a reasonable start, is arbitrary. After-all proper *DAF-7* and serotonin signal regulations depend on everything from gene expression to their secretions as well as additional regulatory steps when their cognate receptors are engaged. Therefore the claim that gene expression should be taken as a direct proxy of increased or reduced activations of these signaling cascades should be toned-down.

6) The authors claim that *daf-7* and *tph-1* define a new class of DR modulators (Results and Discussion) because they mediate a bidirectional lifespan response to DR. However, the other genes they highlight have not been studies in the same manner across multiple food concentrations, so it seems premature to make such a claim. Please rephrase.

7) At the beginning of the Discussion section, the authors state that “we have shown for the first time” that food abundance is functionally encoded in vivo by the combinatorial expression of *tph-1* and *daf-7*. This seems to be more about semantics than a real advance. It was already known that *tph-1* and *daf-7* expressions are food responsive but they had not been analyzed across so many concentrations so carefully.

---

## [Author Response]

*1) One concern pertain to what seem to be modest phenotypic effects of* tph-1 *and* daf-7 *mutations on life span in the presence of food (with preservation of the bidirectional influence of varying food concentrations on life span), a relatively low predictive accuracy of reporter gene expression, and the low magnitude of changes in reporter gene expression observed (both in wild-type animals over the range of food concentrations tested and in* tph-1 *and* daf-7 *mutant animals compared to wild-type animals). These observations lead me to wonder if the authors are focusing on molecules that play an ancillary role in representing food abundance, to the exclusion of undiscovered molecules that are more central to this process. This should be discussed*.

We agree with the comment that *daf-7* and *tph-1* do not constitute the entire response, which we previously pointed out (Results) and which we now mention with greater prominence:

In the Discussion:

“While these results support the idea that *daf-7* and *tph-1* functionally encode significant information […] characterized DR regulators or undiscovered genes.”

Our results support the idea that *daf-7* and *tph-1* encodes a substantial if not the majority of food information. Here, it is important to note the information contained within signalling systems are not only encoded in the response magnitude, but also on dynamic range and variance; all these parameters are regulated by the *daf-7*/*tph-1* circuit. We now emphasize these conclusions from our systematic analysis:

In the Discussion section:

“How much information is conveyed by *daf-7* and *tph-1*? […] This result is particularly remarkable because the predictive accuracy depends on not just the means but also variances.”

Regarding the magnitudes of gene expression and lifespan phenotypes, where comparable experiments have been published, the magnitude of our effects agree with those results, indicating reproducibility. We previously indicated similarities to published DR experiments (Results), and now we also mention similarities under dietary deprivation and in the mutants. Our gene expression responses might appear small for readers accustomed to on/off switches during cell fate specification, but they are consistent with precedents in the literature, such as Figure S4 in Liang et al., Cell Metabolism (2006), and Figure S1 Cunningham et al., Cell Metabolism (2012). Finally, we indicate the predictive accuracy should be considered a lower bound due to experimental noise:

In the Results:

“The longest lifespans occurred in the absence of bacteria, and the magnitudes of these effects are consistent with published dietary deprivation experiments (35; 37).”

“The magnitude of lifespan changes we observed at high food levels (1x10^10^ cells/ml) were comparable to prior studies performed at *ad libitum* food conditions (45; 58).”

In the subsection headed “Decoding Analysis” of the Materials and methods:

“As with all measurements of variance, these accuracy estimates should be considered as lower bounds due to experimental noise.”

*2) The authors repeatedly argued that 5-HT and* DAF-7 *act in parallel to modulate the lifespan. However, a double mutation of* tph-1 *and* daf-7 *did not produce an additive ageing phenotype (*Figure 1*). Please discuss*.

We now elaborate this issue in the third paragraph of the Discussion:

“Our findings provide a more nuanced view of the interactions between the serotoninergic and TGFβ neuroendocrine systems. […] Together, these results point to parallel rather than sequential signalling for serotonin and TGFβ.”

*3) The authors made great efforts to establish differential expression of* tph-1::gfp *in ADF and NSM, particularly when worms grown on very high concentrations of* E. coli *food (*Figures 3, 4 and 5*). While the results are very interesting, the differential* tph-1 *expression does not appear to play a central role in ageing. First, transgenic expression of* tph-1 *in ADF or NSM restored the lifespan as a function of food availability. Second, there is no indication that the expression levels of those transgenes are modulated by food availability. Thus food restriction-induced changes in* tph-1 *expression probably influence other 5-HT signaling pathways but not the ageing processes*.

Our experiments do not indicate rescue of lifespan as a function of food levels by non-food responsive expression of *tph-1* in ADF or NSM. As shown in Figure 2, we only performed *tph-1* rescue experiments on one food level (S Basal) where *tph-1* mutants showed the strongest phenotype. Instead, this result indicates that *tph-1* levels in ADF or NSM matter for lifespan, because lifespan changes when *tph-1* levels are shifted from zero (in the *tph-1* null mutant) to high (in the rescue lines which are overexpressed from multi-copy arrays).

*4) Why do* daf-7 *and* tph-1 *mutants cause extension of lifespan at some food concentrations but reduce it at others? Please discuss*.

We now cover this issue at greater depth, particularly in the Discussion:

“The phenotypes of *tph-1(-)* and *daf-7(-)* mutants reveal that rather than serving intrinsic roles in lifespan extension […] and lowered the accuracy of food level representation (Figure 6).”

*5) The sole reliance on gene expression, while a reasonable start, is arbitrary. After-all proper* DAF-7 *and serotonin signal regulations depend on everything from gene expression to their secretions as well as additional regulatory steps when their cognate receptors are engaged. Therefore the claim that gene expression should be taken as a direct proxy of increased or reduced activations of these signaling cascades should be toned-down*.

We revised our text to indicate that the reporters are faithful readouts for gene expression, rather than pathway activation.

In the Results:

“These published results indicate that *tph-1* and *daf-7* reporters are faithful readouts for the expression of their respective genes (see Methods for additional details on reporter validation).”

Additionally, we used the term “signal” in certain context to mean “gene-expression signal”, since this was potentially confusing, we now clarify that in the text, in the Results section:

“Amplification of dynamic range may contribute to increased encoding capability by increasing separation of gene-expression signals.”

“In contrast to the effects of dynamic range, increasing inter-individual variation degrades encoding capability by increasing overlap between gene expression-signals.”

*6) The authors claim that* daf-7 *and* tph-1 *define a new class of DR modulators (Results and Discussion) because they mediate a bidirectional lifespan response to DR. However, the other genes they highlight have not been studies in the same manner across multiple food concentrations, so it seems premature to make such a claim. Please rephrase*.

We have revised the text to indicate that this is a new phenotypic class and emphasize this comment made by the reviewers:

In the Results:

“Thus, *tph-1* and *daf-7* mutants reveal a previously unobserved DR phenotype, and our results suggest that these genes mediate a bidirectional lifespan response to DR.”

And in the Discussion:

“The phenotypes of *tph-1(-)* and *daf-7(-)* mutants reveal that rather than serving intrinsic roles in lifespan extension, serotonin and TGFβ are signals required for bidirectional food-dependent lifespan changes (Figure 1). This bidirectional phenotype was only revealed by analysis of lifespan across a very broad range of food levels. Many genes have been implicated in lifespan modulation or DR-related responses; a systematic analysis under additional conditions may provide new insights on their roles in food-dependent responses.”

*7) At the beginning of the Discussion section, the authors state that “we have shown for the first time” that food abundance is functionally encoded in vivo by the combinatorial expression of* tph-1 *and* daf-7. *This seems to be more about semantics than a real advance. It was already known that tph-1 and daf-7 expressions are food responsive but they had not been analyzed across so many concentrations so carefully*.

We have revised the text as follows:

“We have shown that food abundance is functionally encoded in vivo in *C. elegans* by the combinatorial expression of *tph-1* and *daf-7* in three neuron pairs.”